# Acoustic shadows help gleaning bats find prey, but may be defeated by prey acoustic camouflage on rough surfaces

Elizabeth L Clare[1,2]*, Marc W Holderied[2]

[1]School of Biological and Chemical Science, Queen Mary University of London, London, United Kingdom; [2]School of Biological Sciences, University of Bristol, Bristol, United Kingdom

**Abstract** Perceptual abilities of animals, like echolocating bats, are difficult to study because they challenge our understanding of non-visual senses. We used novel acoustic tomography to convert echoes into visual representations and compare these cues to traditional echo measurements. We provide a new hypothesis for the echo-acoustic basis of prey detection on surfaces. We propose that bats perceive a change in depth profile and an 'acoustic shadow' cast by prey. The shadow is more salient than prey echoes and particularly strong on smooth surfaces. This may explain why bats look for prey on flat surfaces like leaves using scanning behaviour. We propose that rather than forming search images for prey, whose characteristics are unpredictable, predators may look for disruptions to the resting surface (acoustic shadows). The fact that the acoustic shadow is much fainter on rougher resting surfaces provides the first empirical evidence for 'acoustic camouflage' as an anti-predator defence mechanism.

*For correspondence: e.clare@qmul.ac.uk

Competing interests: The authors declare that no competing interests exist.

## Introduction

Considerable investigative effort has been focused on how flying bats use echolocation to track and capture moving prey in mid-air. Investigations of this aerial hawking behaviour have led to the recognition of novel prey defenses (*Fullard and Napoleone, 2001*; *Ratcliffe and Fullard, 2005*) and methods of prey detection, tracking, and capture (*Goerlitz et al., 2010*; *Conner and Corcoran, 2012*). Less well understood is how bats detect insects and other prey on surfaces. This foraging behaviour (gleaning) is perceptually complex and takes several forms. Some bats exploit sexual advertisement calls of prey to determine their location (e.g., *Myotis septentrionalis* attacks katydids calling for mates on grass tips [*ter Hofstede et al., 2008*], *Trachops cirrhosus* similarly exploits displaying frogs [*Ryan et al., 1982*; *Page and Ryan, 2006*]). Other passive gleaners listen for more subtle prey-generated sounds such as fluttering wings (e.g., rhinolophids and hipposiderids, [*Siemers and Ivanova, 2004*]) and scorpions' walking noises (*Holderied et al., 2011*). Ground-gleaning bats (*Siemers and Ivanova, 2004*) use many of the same strategies of hawking including continuous echolocation and shortened duration of pulse intervals to target moving prey.

Active gleaning, when a bat detects and removes motionless prey from a surface, appears to be the most demanding gleaning behaviour. There is some variation in call structure in gleaning bats. Gleaning *Myotis* use short frequency-modulated (FM) broadband calls, but behavioural evidence demonstrates these bats are less efficient at this task when background 'clutter' obscures target echoes (*Arlettaz et al., 2001*). Some substrate gleaners may beat their wings at potential prey to elicit movement (*Kuc and Kuc, 2012*). True active gleaning requires the bat to distinguish a motionless target from its resting surface, which can be a highly structured background. *Geipel, Jung, and Kalko (2013)* suggested that some species such as *Micronycteris microtis* perform this task expertly.

**eLife digest** While bats are far from blind, they are famed for their use of sound waves to home in on their prey. As they fly, bats send out a series of high-frequency calls that bounce off nearby objects, including insects. By listening to the echoes, the bats are able to build up an auditory image of their environment and thus pinpoint the location of their prey.

Although echolocation is effective for localizing flying insects, it is less suited to detecting those that are resting on surfaces. This is due to the difficulty of distinguishing sound waves that bounce off the insect from those that are reflected by the surrounding surface. While some species of bats get around this problem by listening for faint sounds made by prey, such as mating calls or the fluttering of insect wings, a number of bat species have found a way to detect entirely motionless prey using echolocation.

Clare and Holderied have now worked out how bats might do this. A method was devised to convert the sound waves that bounce off an object into visual signals, and thus, open them up for analysis by human observers. This technique was used to scan moths of various shapes and sizes resting on different surfaces: smooth slate, leaves, coarse limestone, and rough bark. These experiments showed that the difference in the strength of the echoes from the moth and its surroundings varied depending on the texture of the surface.

Specifically, the difference was greatest when the insect was resting on smooth slate and smallest when it was on rough bark, suggesting that choice of surface affects how easy it is to spot an insect. Unexpectedly, however, the data also indicate that bats may search for insects by seeking out interruptions in the echoes from the surface, rather than trying to detect echoes from the prey itself. This makes the bats' task a little easier as it means that they do not have to make adjustments for the differing sizes and shapes of insects. Instead, they can use an acoustic search image for what the surface is like and look for missing parts covered up by the prey.

Clare and Holderied's findings thus generate a number of predictions about the behaviour of bats and insects in their natural environment. Bats should prefer searching for insects on smooth surfaces rather than rough ones; but insects might attempt 'acoustic camouflage' by choosing to rest on rough surfaces rather than smooth.

*M. microtis* uses broad-band FM multi-harmonic calls of very short duration ($\approx$0.2 ms) emitted as single calls with intervals of $\approx$31 ms or in groups of two calls with an intragroup pulse of $\approx$15 ms (*Geipel et al., 2013*). Call start frequencies are $\approx$143 kHz and end frequencies $\approx$69 kHz, and there appear to be no terminal feeding buzzes (*Geipel et al., 2013*). A few authors have used behavioural experiments to investigate the role of approach angles or speculate on the use of a search image (*Jones, 2013*; *Geipel et al., 2013*). Until now, however, no solution to the nature of the underlying acoustic cues has been provided.

Passive gleaning bats use broadband, low-intensity calls (*Faure and Barclay, 1994*), while active gleaners may use much more variable call structures and intensities. Some call characteristics have obvious advantages (e.g., low-intensity calls make bats relatively inaudible to eared prey [*Faure and Barclay, 1994*]), while the importance of other features is unclear, particularly when the surface is highly structured (*Arlettaz et al., 2001*). Previous experiments with gleaning behaviour have either focused on the bats' use of prey-generated sounds or acoustically simple surfaces (*Arlettaz et al., 2001*; *Siemers and Ivanova, 2004*). To our knowledge, no one has considered the specific echo cues used to find motionless prey. In addition, many insects lack ears and/or the wings of resting prey (such as moths) cover the ears, reducing the prey's acoustic sensitivity (*Faure and Barclay, 1994*) and potentially confounding their auditory defences. It is not clear if this resting posture makes them more difficult for bats to detect either by reducing target size or obscuring their body.

We measure conventional single echo cues of root mean square amplitude (RMS) and duration (*Figure 1*) as well as using a novel 'acoustic tomography' technique (*Figure 2*, *Figure 3*) to address two predictions about gleaning. First, we test the prediction that acoustic cues used by predators hunting prey are subtle and that novel cues based on image integration are more salient to the predator. Second, we test the prediction that wing position affects prey echoes. In addition, we consider the effect of differently structured surfaces as these may act to conceal resting targets as

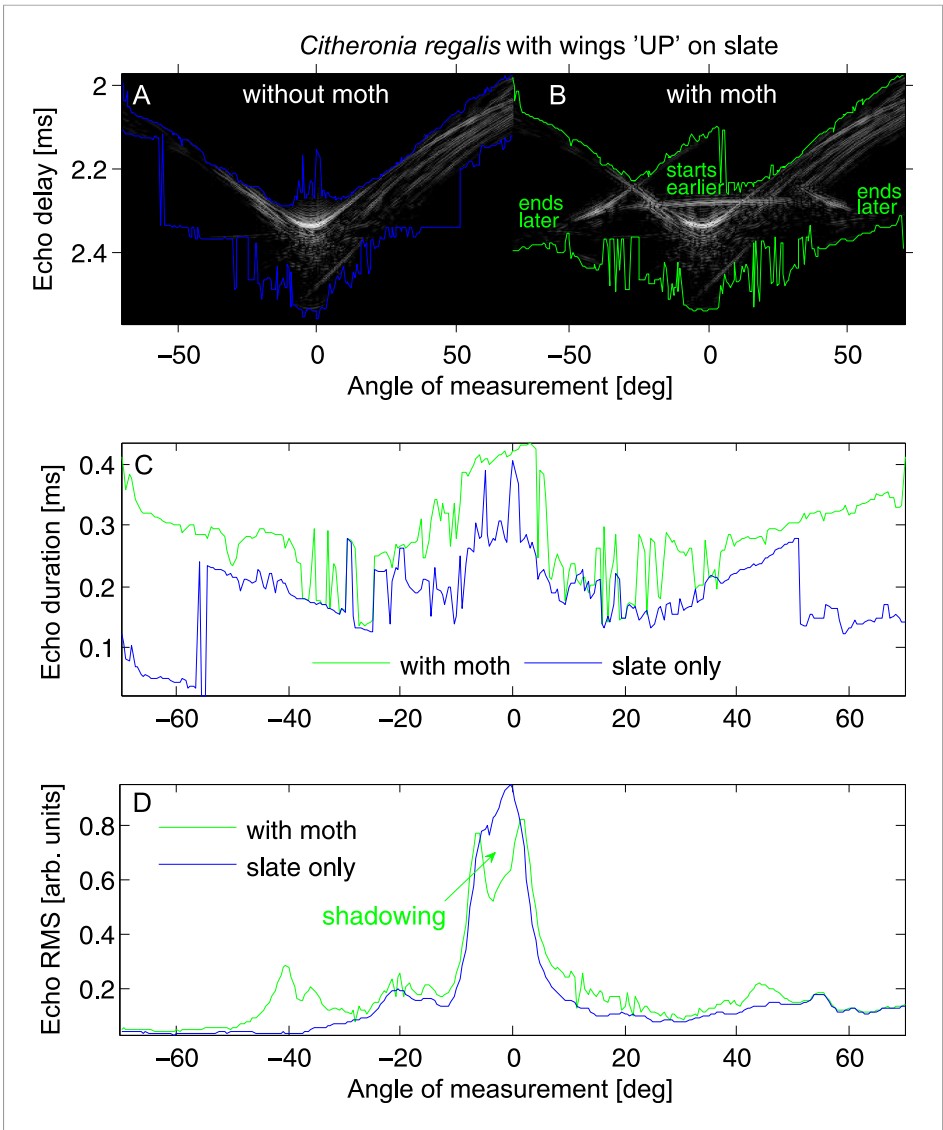

**Figure 1**. Echo cue examples for *Citheronia regalis* on slate with wings in 'UP' position. (**A** and **B**) Envelope of the echo impulse response as a function of measurement angle, (**A**) slate only; (**B**) slate plus moth. Coloured lines indicate start (top lines) and end (bottom lines) of the echo. (**C**) Echo duration and (**D**) echo root mean square (RMS) as a function of measurement angle. Green lines: with moth, blue lines: without moth.

a form of acoustic camouflage, and that consequently there are perceptual benefits to gleaning from acoustically simple surfaces. We also consider the challenge that prey size and shape may vary greatly and be unpredictable. Thus, we include both multiple surfaces and multiple species of prey in our analysis.

Tomographic techniques compile images from a series of sections generated by converting an energy wave into a visual signal. It is used commonly in medical imaging through computed tomography (CT), and acoustic tomography has been applied to some landscape imaging applications (*Duric et al., 2011*). Acoustic tomography in air (*Balleri et al., 2010*) uses CT to interpret sound waves rather than the X-rays more normally associated with medical applications (*Duric et al., 2011*) but has not been widely applied in ecology (*Balleri et al., 2010*). Acoustic impulse responses represent a range profile of echo reflections that have been used to reveal spatial reflection patterns of bat-pollinated flowers and different leaf shapes and determine their role as nectar guides for echolocating bats (*von Helversen and von Helversen, 1999*; *Simon et al., 2011*). Our analysis is

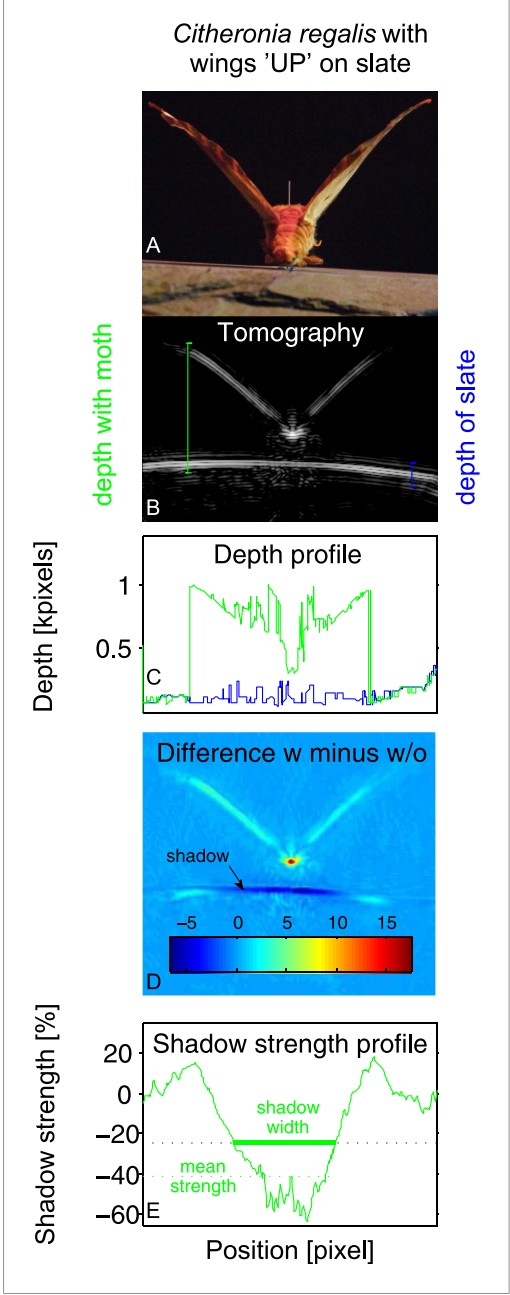

**Figure 2**. Example of tomography analysis for *Citheronia regalis* on slate with wings in 'UP' position.
(**A**) Photograph from rear of moth. (**B**) Tomography. Coloured vertical lines indicate example depth measurements ('with moth' in green; 'without moth' in blue). (**C**) Depth profile for substrate with (green) and without (blue) moth as a function of measurement angle. (**D**) Subtraction of tomographies with (w) and without (w/o) moth. Colour indicates absolute difference. (**E**) Strength of the shadow in percentage (%) difference to without moth as a function of measurement angle. Shadow size is measured as all angles where the shadow strength is at least 25% below the substrate without a moth. Overall shadow strength is the mean shadow strength over the entire shadow.

the first attempt to convert multi-aspect echo-acoustic information into tomographies. Acoustic tomography allows us to transform acoustic information into visual representations. This makes the information-gathering properties of echolocation more conventionally quantifiable giving us access to information not previously accessible. In this study, we use this approach to investigate the perceptual acoustic cues available to active gleaning bats when approaching a motionless prey item on a surface.

## Results

### Substrate cues

*Figure 3E–H* compares the tomographies of the four surfaces: bark, leaf, limestone, and slate revealing that slate presents a flat smooth surface, leaf has a smooth surface disrupted by venation, limestone is relatively flat but with a coarsely granulated surface, and bark presents a highly structured and rough surface. Depth profiles from acoustic tomographies (*Figure 4*) were significantly different among substrates (ANOVA, $F_{3,480} = 335.8$, p < 0.001, Tukey HSD p < 0.001 except limestone and leaf) from a minimal depth profile on slate to a significant profile on bark, thus, we consider the effect of substrate in all further analyses.

### The effect of wing position on tomographies

We found no significant difference in echo-acoustic measurements between wing position using traditional cues (RMS and duration). In the more sensitive tomographies, we compared depth profiles of the substrate generated when a moth was present either with its wings up or down (*Figure 5*). Using a linear mixed-effects model, there was no interaction between wing orientation and substrate ($F_{3,105} = 1.66$, p = 0.18, *Supplementary file 1*; Table 1a for model estimates). Wing orientation predicted depth profiles with the 'UP' position having a greater depth profile on all substrates though the effect here, and with shadow strength, is subtle and due to only a few unusual species (see 'Discussion'). The acoustic shadow cast by the moth was affected by both the wing position and substrate giving a significant interaction in a linear mixed-effects model between the orientation of the wings and the substrate on shadow strength, the 'shadow effect' ($F_{3,105} = 3.19$, p = 0.027, *Supplementary file 1*; Table 1b for model estimates, *Figure 5*).

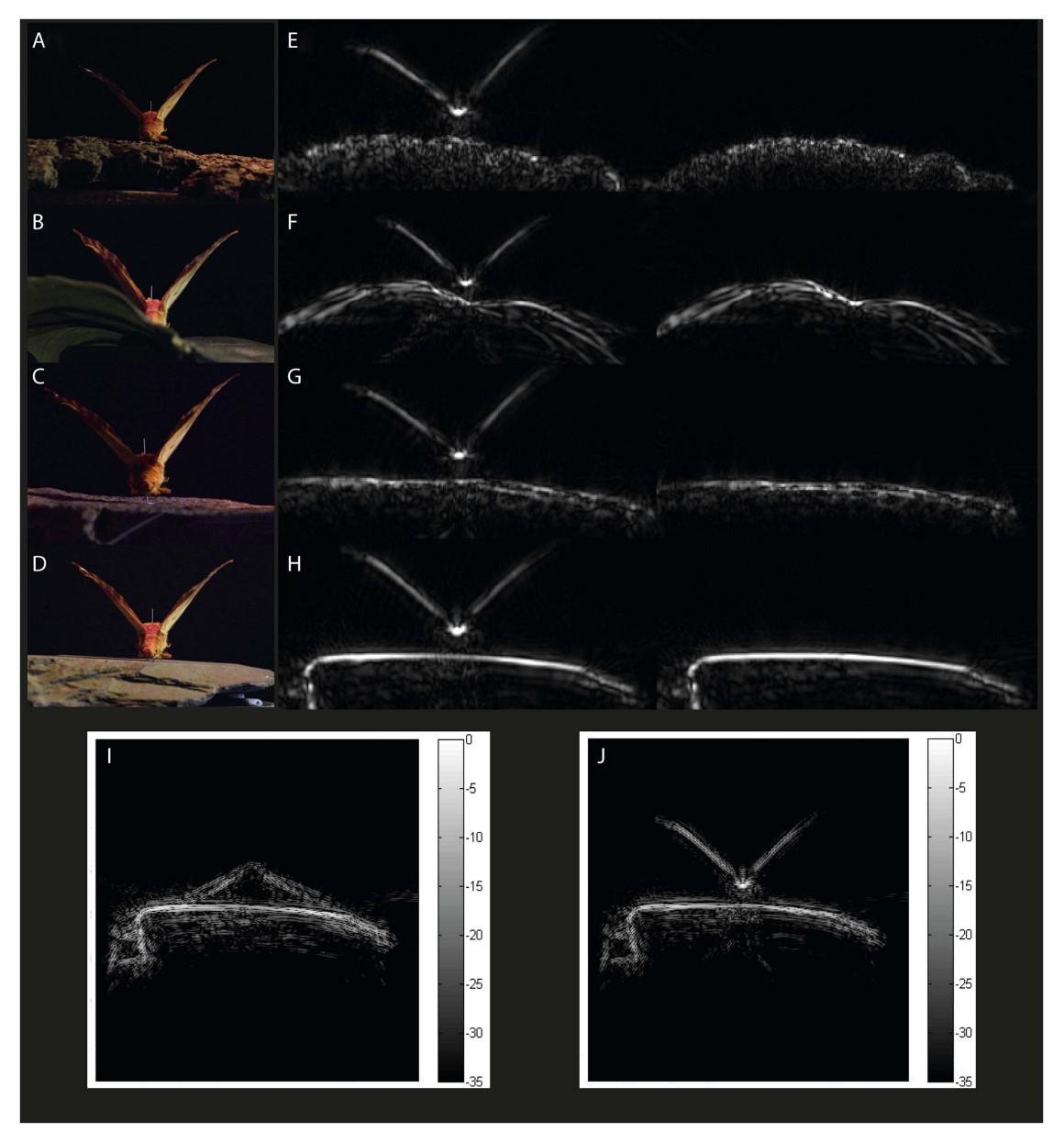

**Figure 3**. Example acoustic tomographies for *Citheronia regalis*. Specimens were placed on one of four substrates (**A**) bark, (**B**) leaf, (**C**) limestone, and (**D**) slate, and we generated acoustic tomographies with and without the specimen (left and right in panels **E**–**H**, respectively). For each species, we generated tomographies with specimens that had their wings (**I**) FLAT and (**J**) UP.

## The effect of substrate and potential acoustic camouflage in tomographies

The most dramatic effect was the choice of moth resting substrate (slate, leaf, limestone, or bark). Substrate roughness itself acts to conceal the moth echo and this effect was particularly important using multi-aspect cues from tomographies. There was a significant decrease in the change in depth profiles on acoustically 'rougher' surfaces (i.e., slate is the most acoustically uniform substrate [*Figure 4*] and generates the greatest depth profile change when the moth was present [*Figure 5A*]). This was supported by our linear mixed-effects model ($F_{3,108} = 108.99$, p < 0001, *Supplementary file 1*;

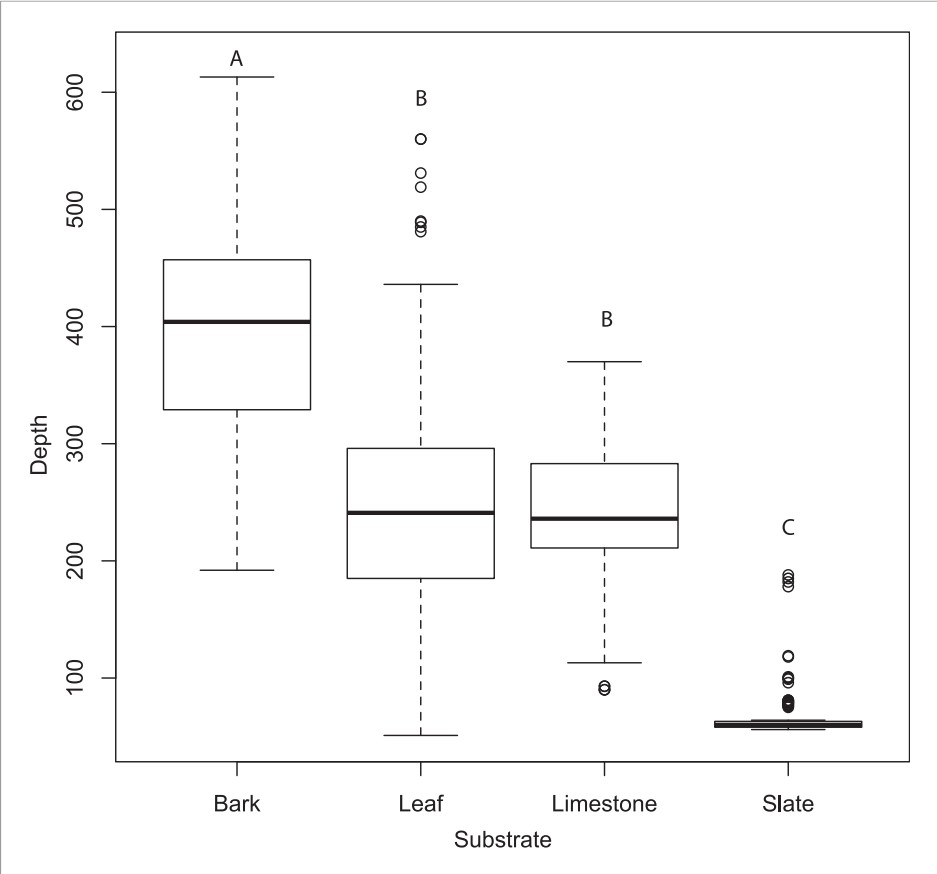

**Figure 4**. Tomographies indicate that substrate predicts mean depth profile with no moth present. Slate presents the most acoustically 'mirror-like' surface with the most minimal depth profile. Letters indicate significant differences. See also *Figure 4—source data 1*.
The following source data are available for figure 4:

**Source data 1**. Depth measures for each substrate.

Table 1c for model estimates, *Figure 5*). Substrate 'roughness' predicts the change in depth profile when a moth is present. The largest overall acoustic shadow effect (regardless of wing position) was seen in the change in shadow strength on leaf and slate substrates where shadow strength was always below 0 (the point where there was no change in acoustic information) with moths casting a strong shadow on the substrate (*Figure 5B*).

## The effect of angle and scanning behaviour on traditional cues

Using traditional single echo cues (log transformed echo duration and RMS amplitude), we found that angle of approach was a predictor of whether traditional echo cues were significantly different with and without a moth. Duration decreased while RMS increased as the angle of approach reached perpendicular direction to the target surface (*Figure 6—figure supplement 1*). We used linear models to assess this effect. The *akaike information criterion* indicated models with interactions were better fitted (*Figure 6*). In a linear model, duration increased with angle on all substrates both without ($F_{3,556} = 40.09$, $p < 0.001$, *Supplementary file 1*; Table 1d for model estimates) and with ($F_{3,556} = 60.3$, $p < 0.001$, *Supplementary file 1*; Table 1e for model estimates) a moth (with the exception of slate without a moth). Leaf behaved most similarly to slate. In a linear model, RMS amplitude decreased on all substrates as angle increased with ($F_{3,556} = 40.48$, $p < 0.001$, *Supplementary file 1*; Table 1f for model estimates) and without ($F_{3,556} = 55.7$, $p < 0.001$, *Supplementary file 1*; Table 1g for model estimates)

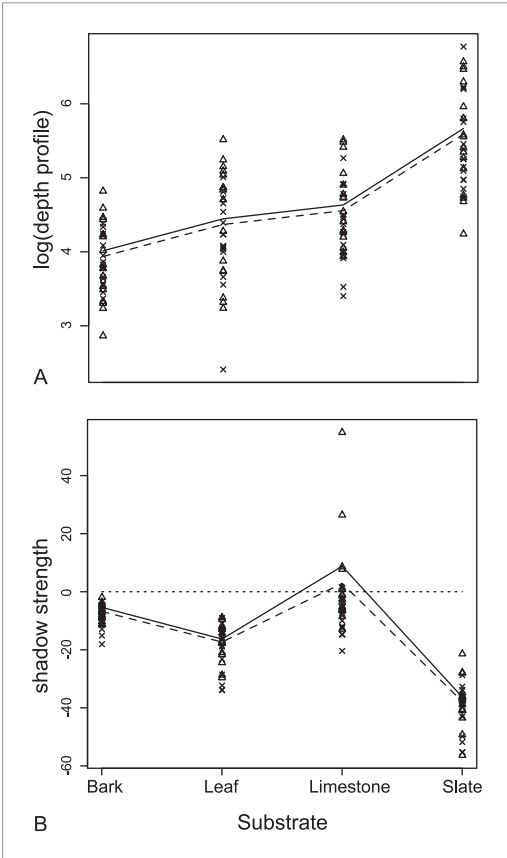

**Figure 5**. Measurements from acoustic tomographies. (**A**) Wing orientation predicted mean depth with the UP position (triangles, solid lines) having a larger depth profile on all substrates than FLAT position (crosses, dashed lines). The biggest effect was of resting substrate with the smooth surface, slate, causing the largest depth profile for each moth. (**B**) There was a significant interaction between the orientation of the wings and the substrate on the measures of shadow strength (missing background). The largest effect of shadow strength was observed on smooth surfaces (leaf and slate). More negative values indicate more obvious missing substrate with zero being no change from substrate alone. See also *Figure 5—source data 1*.

The following source data are available for figure 5:

**Source data 1**. Shadow and depth measurements.

a moth (*Figure 6*) though the decrease was not steady. There were fluctuations in both duration and RMS caused by the venation creating folding of the leaf. In these cases, echo signals are likely jumping between two points of reflection depending on slight changes in angle. The largest magnitude of change in RMS amplitude with angle was seen on leaf.

We used a two factor ANOVA on angles from 0 to 70° averaged in 10° increments to compare the effect of angle of ensonification and moth presence on echo duration and RMS on each substrate. We considered the presence of a moth and angle of ensonification as main effects and explored the interaction to test the hypothesis that the effect of a moth is only significant from certain directions. There was no significant effect of the presence of the moth on RMS at any angle though in almost all cases RMS showed a trend towards higher values when the moth was present (*Figure 6—figure supplement 2*). Angle of ensonification did have a significant main effect on RMS on all substrates ($p < 0.001$ in all models). A Tukey HSD test between angles of ensonification showed specific pairwise differences (*Figure 6—figure supplement 2*). Considering duration, there was a significant interaction between moth and angle on all substrates except limestone, so we treated the comparisons between moth and no-moth separately for each substrate on each angle of ensonification with p-values adjusted for multiple comparisons. At almost all angles of ensonification on all four substrates, duration was longer when a moth was present; however, the effect was only significant in nine of the 28 cases, five of which were on slate. On this substrate, the effect was seen from 20 to 70°, and the magnitude of the effect was particularly large from 50 to 70° (*Figure 6—figure supplement 3*).

## Discussion

Among bats, gleaning behaviour is a less common form of prey detection than hawking with only about a third of species regularly employing this strategy (*Arlettaz et al., 2001*). There is considerable evidence of occasional behavioural flexibility among species (*Ratcliffe and Dawson, 2003*), and the degree of flexibility may be related to brain volume (*Ratcliffe et al., 2006*), however until now, the acoustic basis for gleaning has remained uncertain. Here, we use novel image-integration through tomography to present the first evidence that an acoustic shadow effect may provide the primary cue for prey detection in gleaning. The potential for the use of search images has been suggested (*Jones, 2013*) but until now the nature of the cue has remained mysterious. We further suggest that prey-resting surfaces may disrupt this leading to effective acoustic camouflage and that gleaners may counter this through their choice of hunting surface and the use of search images and acoustic scanning.

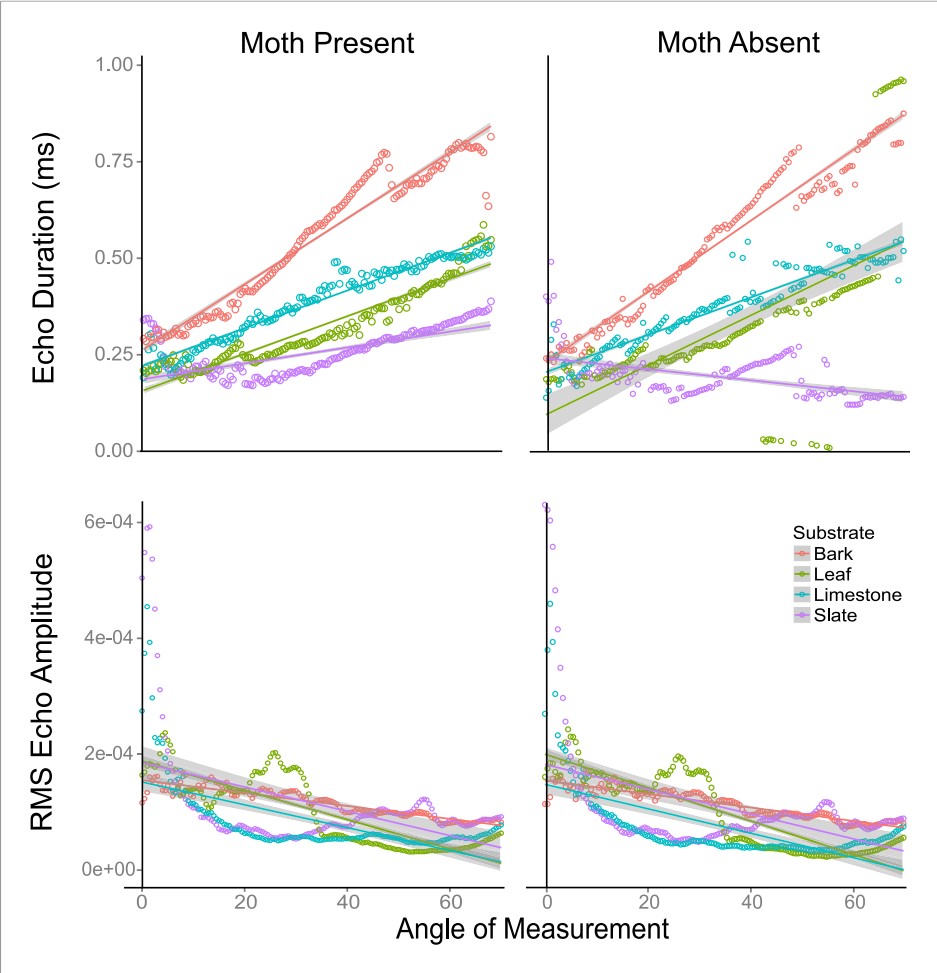

**Figure 6**. Echo duration and RMS echo amplitude log transformed as a function of angles 0–70° relative to the substrate surface for four substrates (slate, leaf, limestone, and bark) with moths present and absent. On all substrates (except leaf without a moth), echo duration decreased towards an angle of incidence that is perpendicular to the substrate surface, with duration higher on substrates that are rougher (bark). Relative RMS echo amplitude increased towards angles perpendicular to the substrate. Smoother, that is, more mirror-like surfaces showed the greatest increase in RMS amplitude for frontal directions. See also *Figure 6—figure supplements 1–5* and *Figure 6—source data 1*.

The following source data and figure supplements are available for figure 6:

**Source data 1**. RMS and Duration measurements.

**Source data 2**. Data associated with *Figure 6—figure supplements 1,4* and **5**.

**Source data 3**. Data associated with *Figure 6—figure supplement 2*.

**Source data 4**. Data associated with *Figure 6—figure supplement 3*.

**Figure supplement 1**. Echo duration (panels **A** and **B**) and RMS echo amplitude (panels **C** and **D**) as a function of angle relative to the substrate surface for four substrates (slate, leaf, limestone, and bark).

**Figure supplement 2**. RMS echo amplitude as a function of 10° angle increments from 1 to 70° to the substrate surface for four substrates (bark, limestone, leaf, slate).

*Figure 6. continued on next page*

*Figure 6. Continued*

**Figure supplement 3**. Duration as a function of 10° angle increments from 1 to 70° to the substrate surface for four substrates (bark, limestone, leaf, slate).

**Figure supplement 4**. Correlations between mean RMS echo amplitude from 1 to 70° when moths were present vs when moths were absent.

**Figure supplement 5**. Correlations between mean duration from 1 to 70° when moths were present vs when moths were absent.

## Gleaning in bats

The low intensity and broadband calls of many gleaning bats differ strongly from the high amplitude and narrowband search calls of aerial hawking species, and it has been demonstrated that one problem with gleaning is an excess of echoes from objects other than the target, referred to as 'echo clutter' (*Arlettaz et al., 2001*). This has led to two alternative explanations for behavioural guilds within gleaning. In the first case, echolocation itself is rarely used because acoustic masking disrupts target echoes making prey-generated sounds more important and most bats functionally passive gleaners (*Arlettaz et al., 2001*). In the second case, broadband calls provide detailed information on texture using spectral cues, which allows for active gleaning in bats to distinguish prey amongst clutter (*Schmidt, 1988*). There has been considerable debate regarding these alternative mechanisms with many authors concluding that alternative signals (prey-generated sounds but also vision and olfaction) are key to prey localization (*Kalko and Condon, 1998*; *Arlettaz et al., 2001*; *Eklöf et al., 2002*); however, some taxa do appear to rely on echolocation alone.

The best example of gleaning in insectivores by echolocation alone was provided in behavioural experiments with the neotropical bat *M. microtis* (*Geipel et al., 2013*). In this case, *M. microtis* was shown to use acoustic cues as the sole sensory modality for prey perception with individuals performing a three-dimensional hovering (scanning) flight in front of prey on leaves while emitting short, multi-harmonic broadband calls. The authors point out that given the short distance between target and substrate, the bats should experience considerable backward masking but they speculate that the hovering in combination with the continual scanning was a key component allowing for ensonification of the target and prey perception by altering the angle between target and background. This might have the effect of reducing clutter and thus the problematic masking (*Geipel et al., 2013*). They further speculate that at obtuse angles, the smooth leaf surface contributes to the 'mirror effect' (*Siemers et al., 2005*), which increases the relative strength of the target echo over the substrate clutter echo, as the latter is mirrored away at oblique angles.

## Tomographic imaging suggests acoustic shadows as a novel cue for gleaning

The power of tomographic imaging is that it provides a representation of the echo source distribution, which lies at the basis of the image perceived by the bat giving us insight into which cues may be most important to the bat rather than an exact measure of the bat's perception. Tomographic imaging is not simply a new tool, but from an information theoretic point of view it provides multi-echo interpretation, which is more powerful in detecting subtle acoustic phenomena.

In our analysis, the most salient cue we measured was the disruption of a surface by the presence of a target prey item revealed in tomographic imaging. This effect, which we dub the 'acoustic shadow', may present fundamental mechanisms allowing bats to find a target on a surface. Interestingly, we find some evidence for acoustic shadowing even in single echo RMS amplitude measurements (*Figure 1D*). Such an acoustic shadow is particularly salient if bats were forming search images (*Jones, 2013*). Our data suggest that scanning behaviour combined with constant patch exploitation and patch fidelity may indicate that true gleaning taxa like *Micronycteris* form a perceptual construct of a predictable hunting environment and then search for disruptions of that construct. This represents a particularly powerful cue in that the bat needs only to perceive a difference in an expected surface, analogous to detecting a prey item resting on a lighter background by a local drop in brightness,

rather than the full features of the prey target whose characteristics depend on size, shape, orientation, and taxon and are therefore unpredictable.

## Bats' perceptual abilities

One important consideration here is the degree of image integration between our experimental system and the biology of the bat. Imaging by sound as we have here gives an indication of which objects create echoes and how strong their overall contribution is. This represents a novel way to measure and interpret the spatial reflection patterns of echo-acoustic energy and is an important new tool in hypothesis generation (finding potentially new cues) and measuring echo-acoustic information in greater detail. However, a bat's image from a single call will be far less detailed (though this cannot currently be quantified). To compensate, bats may use scanning behaviour to change the echo direction slightly between calls and combine multi-aspect range profiles (impulse responses) into a 2D or 3D image through biological tomography analogues. The remaining question is whether a tomographic analogue is biologically plausible, and under realistic conditions, what levels of detail bats are able to achieve. While no behavioural experimentation has yet been conducted specific to our hypothesis, one significant clue to the biological plausibility of a tomographic analogue is the very broadband scanning behaviour of *Micronycteris* (*Geipel et al., 2013*). This behaviour should provide discrimination of depth changes down to 1 mm and combined with the change in approach angle and very short pulse intervals should lead to continuous information during scanning (see an extensive discussion by *Geipel et al. (2013)*).

## The potential for acoustic camouflage and moth resting position

In our analysis, the resting surface significantly affects the potential for successful bat gleaning. Mirror-like surfaces (e.g., leaf or slate) provide more target detection potential across all four acoustic cues we considered. This was subtle for conventional single echo cues and varied with angle but was particularly apparent in multi-aspect integrated image interpretation from tomographies, which provided a far more sensitive measure of acoustic information. This suggests a new phenomenon of acoustic camouflage. Camouflage is most commonly associated with visual concealment with a long history of study and evolutionary interest (e.g., cited as an example in The Origin of Species [*Darwin, 1859*]). Camouflage devices include homochromy, countershading, and disruptive colouration (*Robinson, 1981*) leading to the failure of a predator to detect prey. However, non-visual perceptual forms of camouflage are more rarely considered and are thus harder to define. In our biosonar case, 'rough' surfaces may create echo camouflage for prey making it virtually impossible for bats to differentiate the prey from the unpredictable signals of the surface. This may also constitute 'mimicry' in some cases if the bat detects the prey but dismisses the signal as part of the general variability of the surface (see a discussion of camouflage vs. mimicry in *Robinson (1981)*). Subsequent experimentation with multiple surface gradients is clearly indicated to determine the nature of this effect.

In contrast, a moth's resting position had a very small effect on acoustic cues, mainly due to three specimens. We did not treat these as statistical outliers for biological reasons. Two of these species were hawkmoths (Sphingidae) with very unusual resting positions when wings were up, and the third was a very small butterfly with extremely thin wings. The effect of resting position may be principally hard to detect in our data due to differences in size and this is the main reason for using multiple moth species in our data set. These species were selected to cover a range of potential target sizes and densities (larger moths often have more dense wings and stronger wing venation). We suggest that in most cases, resting position does not provide a meaningful acoustic adaptation. However in some cases, such as hawkmoths, where the body is unusually large compared to the wings, which are folded very tightly when at rest, resting position may be important in detectability.

## The potential impact of surface structure

Our data suggest that acoustically smooth surfaces like leaf and slate will act as mirrors such that among traditional cues at perpendicular directions near zero degrees, RMS will show a peak while duration will be shortest. Across angles from 0 to 70°, these cues (*Figure 1*; *Figure 6—figure supplements 2–5*) showed strong correlation with and without moths present (except duration on slate). At an increasing angle of sound incidence, the echo from the closest parts of the surface arrives

earlier and the most distant parts' echo later, so the overall echo duration increased as the angle moves away to either side (*Figure 1A–C*). The rougher the surface, the more reflectors contribute to the echo and the greater the magnitude of increase in duration (*Figure 6—figure supplement 1*). This suggests that the effect of the moth's presence is subtle and very difficult to detect but that mirror-like surfaces make the task easier by reflecting more information at specific angles yielding more salient cues to the bat. While we did not assess the variability of different surface types (e.g., multiple bark samples or taxa), we suggest that it is the properties of the surface rather than the actual item which matters—thus, smooth surfaces like beech bark may be more acoustically similar to leaves and slate than the bark used here. Further analysis of surface properties is necessary.

## Conclusions

Using the method of acoustic tomography, we have detected a novel acoustic effect, the acoustic shadow, which we present as potentially the most salient cue in gleaning behaviour. Our data also suggest a new effect of acoustic camouflage and the importance of choice in hunting surfaces to overcome this effect for gleaners. Our analysis thus provides a new set of hypotheses regarding the echo-acoustic mechanism of gleaning behaviour and an adaptive explanation for the patch fidelity and leaf scanning behaviour of *M. microtis* (*Geipel et al., 2013*). If the bat is unsure if a prey object is present, scanning maximizes the chance that the prey is ensonified from an angle that returns the most informative single echo cues (*Siemers et al., 2005*) making prey detection more likely, but this scanning also would allow multi-aspect integration potentially leading to biological tomography analogues.

We provide clear evidence for the role that search images may play and thus a strong hypothesis that these are key to gleaning but, contrary to previous speculation (*Jones, 2013*), we suggest the search image is not necessarily for prey, but for surfaces. These effects on the surface image are most salient on mirror-like surfaces. Rough surfaces provide little information even in the most salient acoustic shadow cue. Thus, insects resting on these surfaces are effectively camouflaged by the acoustic roughness of the substrate.

## Materials and methods

### Specimens, substrates, and experimental design

To accommodate potential target variability, we selected two pinned moth specimens from each of 16 species, which varied strongly in size and shape (*Sphinx kalmiae, Sphecodina abbottii, Paonias myops, Dryocampa rubicunda, Citheronia regalis, Erebia pawloskii, Boloria chariclea, Macrurocampa marthesia, Catocala ilia, Ennomos magnaria, Campaea perlata, Lophocampa maculata, Haploa confusa, Grammia virguncula, Grammia virgo, Panopoda rufimargo*) with their wings positioned upwards (UP) in a V in line with their bodies (like many diurnal butterflies) or flat/downwards (FLAT) roughly in the plane of the substrate surface (like many moths) (*Figure 3*). We placed each specimen on four substrates (slate, artificial leaf (leaf), limestone, and rough bark; *Figure 3*) for tomography ($16 \times 2 \times 4 = 128$ images). For each combination of moth, wing position, and substrate, echoes were taken (details see below) from 281 directions, and from the echo impulse responses, we measured four acoustic cues. Two are traditional single echo measures (duration and RMS amplitude), while two are derived from tomographies, which integrate multiple echoes (depth and shadow strength). For each echo, we calculated echo duration (time from the start of the echo until the end of the echo as determined from the envelope of the impulse response with a common constant threshold manually set to be above the noise floor; compare *Figure 1A,B*), and—as measure of acoustic energy—RMS amplitude (calculated over the time period from 1.64 ms to 3.0 ms echo delay; compare *Figure 1A,B*) with and without moths present to quantify the effect of measurement angle on acoustic information. We turned each set of 281 impulse responses into an acoustic tomography using inverse radon transformation (*Balleri et al., 2010*). A tomography is a scaled 2-dimensional acoustic representation of spatial echo origin and strength in a cross section through the target object in the plane echo measurements were taken from. Using tomographies, we can investigate changes to the acoustic profiles of the substrates by the presence of the prey item. We considered two variables, the mean depth profile and the acoustic shadow strength cast by prey on the substrate (compare *Figure 2*; details see below).

## Acoustic tomography

We used a custom-made acoustic tomography system (*Balleri et al., 2010*), consisting of a 1/4″ ultrasound microphone (type 40BF), pre-amplifier (type 26AB; both G.R.A.S Sound & Vibration, Holte, Denmark), 2200C amplifier (Larson Davis Inc., Depew, NY; gain +40 dB), and a custom-made ferro-electret foil loudspeaker (Emfit Ltd., Vaajakoski, Finland) driven by a PZD350 M/S high-voltage amplifier (TREK Inc., Lockport, NY). Microphone and speaker pointed in the same direction and were positioned at a distance of 20 mm from centre to centre to simulate the arrangement of a bat's mouth and ear. Loudspeaker and microphone were mounted on an adjustable lever arm moved by a LT360 turntable (LinearX systems Inc., Battle Ground, WA). Microphone, loudspeaker, and turntable were connected to a NI-DAQ BNC-2110 card operated through LabView v.8.0 (both National Instruments, Austin, TX) with custom-programmes MisureBinauralaverage.vi (rev 337) and FinalProgramBinaural.vi (rev 185) (*Balleri et al., 2010*).

Acoustic measurements were taken in a 2 × 2.3 × 4 m semi-anechoic room. We played linear frequency-modulated sweeps from 250 to 10 kHz of 10-ms duration at 500 kHz sampling rate and 16-bit resolution and recorded sample-synchronously at the same rate and resolution. The echo impulse response was calculated by pulse-forming through deconvolution with the echo recorded perpendicularly from a 60 × 60 cm metal plate. Envelopes were calculated using the absolute value of the Hilbert transform of the impulse response. Echoes of each target were taken from a distance of 100 cm from the target from 281 radial positions (0.5° steps over ±70° relative to perpendicular to the substrate surface).

We used MatLab (v7.5, MathWorks, Natick, MA) to turn sets of impulse responses into acoustic tomographies (*Balleri et al., 2010*). We imposed a colour scale in each tomographic image scaled to amplitude where pixel colour is an indicator of acoustic energy (*Figure 2*).

## Acoustic characters measured from tomographies

In these cross section-like tomographic images, we then defined areas of interest, one including the entire image of the moth including the image of the substrate below, and the other just including the substrate below the moth. From these areas, we then derived two measures of echo-acoustic salience. First, we calculated the change in depth profile from the tomography as the range of pixels starting with the first pixel above background noise (either moth or substrate echo) to the last such pixel of substrate echo in a perpendicular direction relative to the substrate surface (see *Figure 2B*), to derive a detailed depth profile, which builds the informational basis of the bats' perception of an object projecting off the surface (see *Figure 2C*). Second, we measured 'shadow size' and 'shadow strength'. Shadow size and strength were determined by first subtracting the tomography without a moth from the equivalent tomography with the moth (see *Figure 2D*). For each column of pixels (horizontal position), we then measured how much darker (in percent of pixel brightness, i.e., acoustic energy) the tomographic image of the substrate got through the shadow cast by the moth (shadow strength profile; *Figure 2E*). From this shadow strength profile, we then defined the shadow area as all horizontal positions when the substrate echo had fallen by at least 25% and measured 'shadow size' from the first to the last such pixel. 'Shadow strength' was then measured as the mean percentage value of the shadow strength profile over the entire shadow area (see *Figure 2E*).

For each specimen, we scaled the measurements' area of interest to specimen size by calculating the respective shadow size. To avoid bias introduced by different echo clarity on different substrates, we measured shadow size cast by each specimen from slate, because this substrate gave the most homogeneous substrate echoes and thus the most clearly defined shadows (*Figure 2*). Subsequently, we applied this same shadow size to all other substrates but centred the defined area by eye with the visible moth to account for slight variations (<2 mm) in moth pin position.

## Acknowledgements

We thank Tom Mansfield and Gwen Jenkins who conducted initial testing of this experiment during the course of their undergraduate projects with us. We thank Prof James Simmons for previous comments on the analysis and Drs Rob Knell and Steve Le Comber for statistical advice. We thank the Biodiversity Institute of Ontario (University of Guelph, Ontario Canada) for specimen loan. Three reviewers provided substantial review suggestions that greatly improved the quality of this manuscript.

# Additional information

## Funding

| Funder | Grant reference | Author |
| --- | --- | --- |
| Natural Sciences and Engineering Research Council of Canada | Post Doctoral Fellowship | Elizabeth L Clare |

The funder had no role in study design, data collection and interpretation, or the decision to submit the work for publication.

## Author contributions

ELC, MWH, Conception and design, Acquisition of data, Analysis and interpretation of data, Drafting or revising the article

# Additional files

## Supplementary file

• Supplementary file 1. Additional statistical details referring to all tests.

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
