## [Decision Letter]

Thank you for sending your work entitled “Resting substrates provide acoustic camouflage for moths preyed upon by gleaning bats” for consideration at *eLife*. Your article has been favorably evaluated by Ian Baldwin (Senior Editor) and three reviewers, one of whom is a member of our Board of Reviewing Editors. One of the three reviewers, Brock Fenton, has agreed to share his identity.

The reviewers have discussed their reviews with one another, and the Reviewing Editor has drafted this decision to help you prepare a revised submission.

All agree that this work has the potential to be an important contribution to the field, however, it was considered that while the hypothesis being presented is very interesting, more work needs to be done to relate this to the behavior of bats and to improve the clarity of the presentation. It is still unclear from the results presented how bats would actually use such information. The suggestion to build and test a classifier would help considerably to relate this more closely to the biology of bats.

Reviewer #1:

The paper titled: “Resting substrates provide acoustic camouflage for moths preyed upon by gleaning bats” presents a new exciting tool to analyze echoes (acoustic tomography) and it uses this tool to address an interesting riddle – how do gleaning bats detect prey that is stationary on its background? However, I feel that many adjustments must be made in order to make this manuscript the high impact paper it should be:

1) Lack of variance – it was not clear to me from the Methods how variance is achieved in the recordings. It seems like the authors were using the different moth species as repeats for the 'with moth' condition which is ok, and actually even more difficult than using many specimens of the same moth. But what was done to measure many different backgrounds? Were many 'pieces' of background measured for each substrate? Without such variance it is very difficult to run a fair comparison of the 'with' vs. 'without' moth conditions which is what this paper tries to do. Perhaps the variance of different specimens of the same substrate is much larger than the difference between the 'with' and 'without' moth conditions?

2) I would change the order of the Results. Currently the reader must go through several paragraphs describing less interesting and often expected results (e.g. RMS increases with angle), while the main point of the paper is only reached in the last paragraphs. I think that the heart of this paper is the new insight provided by acoustic tomography. This is also what takes up 90% of the Discussion. I would therefore write the manuscript around the tomography results.

3) The authors constantly use the term “detection,” although to my best understanding they never ran any test for detection. They only statistically compare the 'with' and 'without' moth conditions. In order to report detection they should have run a classifier that receives an echo blindly and classifies it as returning from the 'with' or 'without' moth condition. I think that such a test would improve the manuscript significantly, but if it is not added, the terminology at least should change.

4) In the Methods, the part describing the tomography measures (depth and shadow) are very hard to understand and these are the most important parameters for the paper (and the less familiar parameters). I think that this part should be rephrased. The Figure (currently 5) should be better used – the authors could add a depiction of depth and perhaps equations would make things easier. This is not only a formulation concern. Without a complete understanding of these parameters and how they were assessed, it is hard to understand the power of the results. For example, the normalization used by the authors is not fully clear to me and it is not clear if a bat could perform such normalization. If the authors chose to use different moth species to test the (more difficult) case of a bat detecting any moth, then normalizing for size seems to eliminate this variability. Another unclear point is, how could a bat use the shadow parameter? It seems to me that it needs a reference of the surface without the moth (and this again raises the lack of variance mentioned above).

Reviewer #2:

This is an interesting, timely and thoughtful contribution. The authors have shown considerable innovation in their approach to the topic. There are, however, several points that would benefit from further amplification:

1) The echolocation calls of gleaning bats. Important components include frequencies in calls, intensities of calls, durations of calls. Possible concerns are insects hearing and responding to calls, and bat being bombarded by returning echoes.

2) The issue of feeding buzzes during gleaning attacks needs some attention.

3) I did not find the figures easy to understand. Perhaps, in an ideal world, the first figure would show a standard digital image insect on background, then the tomographic view highlighting the salient points and features. There are several figures, but they need some more thought. The pictures will tell the story; spend more time setting them up.

4) Finish with some specific predictions about how bats behave and how prey ‘conceal’ themselves. I recall a paper about behaviour against visually hunting predators; that would be a good model.

Reviewer #3:

1) I find this a useful work that employs new technology to provide a very plausible hypothesis, the ‘missing background’ effect, for how substrate gleaning bats may detect prey. The acoustic tomography approach developed here will likely be of broad importance in the study of echolocation. However, the paper predominantly introduces, rather than tests, these hypotheses and this should be acknowledged more clearly.

2) Furthermore, there are issues with regard to the biological plausibility as it is not clear to me whether bats would, under realistic conditions, be able to obtain the resolution employed in the tomographic technique. As the authors themselves write in the Discussion, “[…] a bat's image from a single call will be far less detailed”. I assume, then, that bats must employ multiple calls for this hypothesis to be plausible, but since we don't know how “less detailed” each call would be, we don't know if this technique represents the bat's world appropriately or not. This is a major issue that needs to be resolved.

3) I think the paper will also benefit from some restructuring. It would seem that Figure 4 should come first as it sets the scene for the study and provides a compelling visual of the technique. Indeed, the authors themselves cite Figure 4 first in their Results, making it seem particularly strange for it to be placed as it is. Following Figure 4, Figure 1 is referenced and then Figure 3. The Results are relatively straightforward. I felt more could be done to determine the plausibility of “detection.” That repeated tests are statistically significant is not as convincing as employing a classifier to determine whether a month is 'perceived' as present or absent. This would also help the reader better understand the key issue here, which is how bats may determine whether a background is “missing” or not.

4) What is somewhat confusing in the manuscript is the use of multiple species, which only became evident to me in the Discussion when discussing hawkmoths and a very small butterfly used. In the statistical tests, were each species compared for presence and absence separately? This makes me feel even more strongly that the use of a classifier would be more beneficial than the statistical tests performed. The experimental use of multiple species needs to be made much more clear in the Introduction and the start of the Results.

5) How does the resolution of the images obtained compare with the estimated “resolution” available to bats? The authors write that “[…] a bat’s image from a single call will be far less detailed”. How much “less detailed”? How many calls would be required to approximate this level of detail? Why was this resolution of tomography employed? We need to be given more detailed and quantitative information as to the biological plausibility of this approach. I very much like the idea of multi-echo interpretation but whether this equipment can detect months is very different to whether bats under realistic scenarios could employ this technique.

6) The work has potential, but this reader is left wondering whether this does or does not provide evidence that bats may actually use this tomographic approach. I think that more clearly defining the relation between the resolution of the man-made tomography with that possible by bats under reasonable conditions would help a lot. It would, for example, be relatively straightforward to downsample the tomographic image to inform us how decreasing resolution impacts detection. In addition, including a classifier approach would make this more convincing and, as well as relating to the range of resolutions available to real organisms, would provide more direct evidence of the biological plausibility of this interesting mechanism.

---

## [Author Response]

Reviewer #1:

*The paper titled:* “*Resting substrates provide acoustic camouflage for moths preyed upon by gleaning bats*” *presents a new exciting tool to analyze echoes (acoustic tomography) and it uses this tool to address an interesting riddle – how do gleaning bats detect prey that is stationary on its background. However, I feel that many adjustments must be made in order to make this manuscript the high impact paper it should be*.

We have done our best to extensively revise the manuscript as suggested. We feel the story and manuscript text is now tighter and the figure more illustrative of the new tomography method.

1) Lack of variance – it was not clear to me from the Methods how variance is achieved in the recordings. It seems like the authors were using the different moth species as repeats for the 'with moth' condition which is ok, and actually even more difficult than using many specimens of the same moth. But what was done to measure many different backgrounds? Were many 'pieces' of background measured for each substrate? Without such variance it is very difficult to run a fair comparison of the 'with' vs. 'without' moth conditions which is what this paper tries to do. Perhaps the variance of different specimens of the same substrate is much larger than the difference between the 'with' and 'without' moth conditions?

Variance was generated by making multiple measurements across multiple angles on multiple specimens of multiple species paired against blank surfaces for four separate cues – amounting to thousands of independent measurements. Some of these were averaged (e.g. tomography) while some are used to consider alternative independent effects (e.g. the effect of angle).

Our initial hypothesis was based around whether acoustic tomography can be used to find new acoustic cues and whether wing position mattered to moths. We used multiple prey species because the size of prey and the thickness of wings etc. are non-trivial and unpredictable problems for a bat. We have now added that discussion to the manuscript. We initially used different substrates simply to see if our results were consistent on different backgrounds. We did not use multiple backgrounds of the same type, as it was not a hypothesis we intended to pursue. During statistical analysis, we determined that background mattered a great deal and we thus considered it a separate variable in a random effects model – we cannot lump these results together but must deal with them as separate conditions. However, we agree that there is minimal variation within substrate (other than measuring from a multiple of angles) and a subsequent study should consider this new dimension. We have altered our manuscript accordingly including the title, hypotheses and predictions to emphasize the correct components and we comment on the effect of roughness and smoothness as a potential factor in the results. A subsequent project should consider the degree to which backgrounds are variable, but that is beyond our analytical scope here.

*2) I would change the order of the Results. Currently the reader must go through several paragraphs describing less interesting and often expected results (e.g. RMS increases with angle), while the main point of the paper is only reached in the last paragraphs. I think that the heart of this paper is the new insight provided by acoustic tomography. This is also what takes up 90% of the Discussion. I would therefore write the manuscript around the tomography results*.

We agree and we have done this.

*3) The authors constantly use the term “detection,” although to my best understanding they never ran any test for detection. They only statistically compare the 'with' and 'without' moth conditions. In order to report detection they should have run a classifier that receives an echo blindly and classifies it as returning from the 'with' or 'without' moth condition. I think that such a test would improve the manuscript significantly, but if it is not added, the terminology at least should change*.

We have considered this and consulted with colleagues. We have also sought clarification from the editor on what a “classifier” would be. We have not had a satisfactory answer to this problem and two statisticians we consulted were unsure what subsequent analysis beyond a statistical outcome would be needed.

As the reviewer suggests, and as was suggested when we consulted the editor, we suspect this is a problem of wording in our results. We simply meant to indicate a difference was significant. If the reviewer means some sort of subsequent statistical analysis based on a bat’s perceptual ability beyond a test of significance – we do not think any such information exists, particularly for gleaning bats. We would need to use behavioral experiments to determine what magnitude of change they interpret as meaningful before we can classify any individual measurement.

To correct this problem we have attempted to change the terminology throughout to remove the word “detection” which was not our intent. We hope our intent is now clearer.

*4) In the Methods, the part describing the tomography measures (depth and shadow) are very hard to understand and these are the most important parameters for the paper (and the less familiar parameters). I think that this part should be rephrased. The Figure (currently 5) should be better used – the authors could add a depiction of depth and perhaps equations would make things easier. This is not only a formulation concern. Without a complete understanding of these parameters and how they were assessed, it is hard to understand the power of the results. For example, the normalization used by the authors is not fully clear to me and it is not clear if a bat could perform such normalization. If the authors chose to use different moth species to test the (more difficult) case of a bat detecting any moth, then normalizing for size seems to eliminate this variability. Another unclear point is, how could a bat use the shadow parameter? It seems to me that it needs a reference of the surface without the moth (and this again raises the lack of variance mentioned above)*.

We completely agree and have changed the figures and Methods section to incorporate more detail and to show one detailed example of measurements; in this case, one moth taken through each measurement (as suggested by another reviewer). In parallel, we have largely reworded the Methods and linked them to the new figure. We also add a new figure serving an analogous function for non-tomographic cues to clarify them. We also add a discussion on the potential use of the shadow cue by bats and the biological analogue. We hope it is now clear that we predict it is used in conjunction with a search image.

Reviewer #2:

*This is an interesting, timely and thoughtful contribution. The authors have shown considerable innovation in their approach to the topic. There are, however, several points that would benefit from further amplification*:

*1) The echolocation calls of gleaning bats. Important components include frequencies in calls, intensities of calls, durations of calls. Possible concerns are insects hearing and responding to calls, and bat being bombarded by returning echoes*.

We have added the description of these calls as reported for *Micronycters* – the best example of this behavior known.

*2) The issue of feeding buzzes during gleaning attacks needs some attention*.

To our knowledge, no feeding buzz during a gleaning attack has been reported. We have added this. There is a grey area of behavior, for example hunting over water where buzzes may be used, but the complexity of those cues are beyond the scope of this manuscript and the behaviour has been well described in other places. Thus we have confined our discussion to terrestrial gleaning to avoid considerable confusion.

*3) I did not find the figures easy to understand. Perhaps, in an ideal world, the first figure would show a standard digital image insect on background, then the tomographic view highlighting the salient points and features. There are several figures, but they need some more thought. The pictures will tell the story; spend more time setting them up*.

We have completely restructured our Methods and redesigned the respective figures accordingly. In this we follow exactly the suggestions of this reviewer and Reviewer 1, where one insect is taken through all measurements. We hope this is now clear.

*4) Finish with some specific predictions about how bats behave and how prey ‘conceal’ themselves. I recall a paper about behaviour against visually hunting predators; that would be a good model*.

We have added a section on perception by biosonar and in particular by surface templates and also add a specific section on potential biosonar camouflage on certain substrates.

Reviewer #3:

*1) I find this a useful work that employs new technology to provide a very plausible hypothesis, the ‘missing background’ effect, for how substrate gleaning bats may detect prey. The acoustic tomography approach developed here will likely be of broad importance in the study of echolocation. However, the paper predominantly introduces, rather than tests, these hypotheses and this should be acknowledged more clearly*.

We have done this. In several places we have now specifically stated the hypothesis generating component of this work. For example the conclusion now incorporates this.

*2) Furthermore, there are issues with regard to the biological plausibility as it is not clear to me whether bats would, under realistic conditions, be able to obtain the resolution employed in the tomographic technique. As the authors themselves write in the Discussion,* “*[…] a bat's image from a single call will be far less detailed*”*. I assume, then, that bats must employ multiple calls for this hypothesis to be plausible, but since we don't know how “less detailed” each call would be, we don't know if this technique represents the bat's world appropriately or not. This is a major issue that needs to be resolved*.

We have added a subsection in the Discussion about this. To a large extent, we can only speculate. The calls themselves would not be less detailed, it is unclear if they can use as many calls. The number of behavioral experiments in gleaning is small, and among active gleaning specialist insectivores that number is very tiny indeed. It is an area that requires considerable investigation, particularly in relation to the way we use similar radar technologies. Since we are introducing an entirely new potential cue, we can only really speculate about what is biologically meaningful. We have attempted to draw on the recorded behavior of *Micronycteris* to demonstrate how this could work, but clearly a behavioural experiment would be needed to attempt to validate this. It is an excellent next field step in the examination of these effects.

*3) I think the paper will also benefit from some restructuring. It would seem that*
Figure 4
*should come first as it sets the scene for the study and provides a compelling visual of the technique. Indeed, the authors themselves cite*
Figure 4
*first in their Results, making it seem particularly strange for it to be placed as it is. Following*
Figure 4*,*
Figure 1
*is referenced and then*
Figure 3*. The Results are relatively straightforward. I felt more could be done to determine the plausibility of* “*detection.*” *That repeated tests are statistically significant is not as convincing as employing a classifier to determine whether a month is 'perceived' as present or absent. This would also help the reader better understand the key issue here, which is how bats may determine whether a background is* “*missing*” *or not*.

We have substantially rewritten the manuscript and reordered the figures. We have added a new figure as per Reviewer 2’s requests for clarification on the methods. Please see our response to reviewer one about a classifier. Despite considerable inquiry, we are not clear what a classifier would be or how any subsequent analysis can be made without a behavioural experiment to determine what magnitude of effects a bat can use. We have revised to remove this ambiguity from the text.

*4) What is somewhat confusing in the manuscript is the use of multiple species, which only became evident to me in the Discussion when discussing hawkmoths and a very small butterfly used. In the statistical tests, were each species compared for presence and absence separately? This makes me feel even more strongly that the use of a classifier would be more beneficial than the statistical tests performed. The experimental use of multiple species needs to be made much more clear in the Introduction and the start of the Results*.

We clearly described the use of multiple species in the opening of the Methods section, but we agree that the order of sections required in *eLife* does end up burying this at the bottom of the paper. We have now added it in a couple of other places with some clarification. We used multiple species in a complete paired design as the reviewer suggested (no missing data). The purpose of the multiple species is because variations in size and shape are a very real and very unpredictable problem for a bat. Without including this we could not generalize any of the conclusions beyond the effect one a single prey item. We hoped the use of a wide variety of species would generate the variability that is biologically meaningful.

*5) How does the resolution of the images obtained compare with the estimated* “*resolution*” *available to bats? The authors write that* “*[…] a bat’s image from a single call will be far less detailed*”*. How much “less detailed”? How many calls would be required to approximate this level of detail? Why was this resolution of tomography employed? We need to be given more detailed and quantitative information as to the biological plausibility of this approach. I very much like the idea of multi-echo interpretation but whether this equipment can detect months is very different to whether bats under realistic scenarios could employ this technique*.

This is similar to the question raised above (the reviewer uses the same quotation) about what is biologically meaningful. It is essentially a very old question: what does the bat perceive? This is very different from: what are bats capable of discriminating? This latter component is what is quite often investigated. For example, a good echolocating bat can distinguish two points only a few millimeters apart but may still attack a stone in the air as if it is prey. Clearly, “capable” is not the same as “meaningful”. Tomography might help solve this problem by giving us a more integrated idea of what echolocation can do. However, we will need very extensive behavioural experiments to make the leap to perception. We have added a new section about this problem and have drawn on the behavior of *Micronycteris* to provide a potential explanation of the biological analogue.

*6) The work has potential, but this reader is left wondering whether this does or does not provide evidence that bats may actually use this tomographic approach. I think that more clearly defining the relation between the resolution of the man-made tomography with that possible by bats under reasonable conditions would help a lot. It would, for example, be relatively straightforward to downsample the tomographic image to inform us how decreasing resolution impacts detection. In addition, including a classifier approach would make this more convincing and, as well as relating to the range of resolutions available to real organisms, would provide more direct evidence of the biological plausibility of this interesting mechanism*.

Our attempt here is to present a new mechanism for biological testing not previously available because of technological limitations. Downscaling, as suggested, will only be useful when we know more about bat perception – and so is not going to answer the question presented yet. One reason to include the more standard cues (RMS, duration) is to compare a very simplified view to our tomographic information.